# LEVERAGING INDUCTIVE BIAS OF NEURAL NETWORKS FOR LEARNING WITHOUT EXPLICIT HUMAN ANNOTATIONS

## ABSTRACT

Image classification problems today are often solved by first collecting examples along with candidate labels, second obtaining clean labels from workers, and third training a large, overparameterized deep neural network on the clean examples. The second, manual labeling step is often the most expensive one as it requires manually going through all examples. In this paper we propose to i) skip the manual labeling step entirely, ii) directly train the deep neural network on the noisy candidate labels, and iii) early stop the training to avoid overfitting. With this procedure we exploit an intriguing property of overparameterized neural networks: While they are capable of perfectly fitting the noisy data, gradient descent fits clean labels faster than noisy ones. Thus, training and early stopping on noisy labels resembles training on clean labels only. Our results show that early stopping the training of standard deep networks (such as ResNet-18) on a subset of the Tiny Images dataset (which is obtained without any explicit human labels and only about half of the labels are correct), gives a significantly higher test performance than when trained on the clean CIFAR-10 training dataset (which is obtained by labeling a subset of the Tiny Images dataset). In addition, our results show that the noise generated through the label collection process is not nearly as adversarial for learning as the noise generated by randomly flipping labels, which is the noise most prevalent in works demonstrating noise robustness of neural networks.

## 1 INTRODUCTION

Much of the recent success in building machine learning systems for image classification can be attributed to training deep neural networks on large, humanly annotated datasets, such as ImageNet (Deng et al., 2009) or CIFAR-10 (Torralba et al., 2008). However, assembling such datasets is time-consuming and expensive: Both ImageNet and CIFAR-10 were constructed by first searching the web for candidate images and second labeling the candidate images by workers to obtain clean labels instead of the noisy candidate ones. The first step already yields labeled examples, but the accuracy of those automatically collected, candidate labels is low: For example, only about half of the labels of the Tiny Images dataset constructed in the first step of obtaining the CIFAR-10 dataset are correct (Torralba et al., 2008). In the second step, clean labels are obtained by asking workers through a crowdsourcing platform for annotations and aggregating them.

In this paper we propose to train directly on the noisy candidate examples, effectively skipping the expensive human labeling step. To make this work, we exploit an intriguing property of large, overparameterized neural networks: If trained with stochastic gradient descent or variants thereof, neural networks fit clean labels significantly faster than noisy ones. This fact is well known, see for example the experiments by Zhang et al. (2017, Fig. 1a) which demonstrated that overparameterized deep networks can fit even randomly shuffled CIFAR-10 labels perfectly. What is now well known is that this effect is sufficiently strong to enable training on candidate labels only. Our idea is that, if neural networks fit clean labels faster than noise, then training them on a set containing clean and wrong labels and early stopping the training resembles training on the clean labels only.

## 1.1 OUR CONTRIBUTIONS

We show that early stopping the training on candidate examples can achieve classification performance higher than when training on clean labels, provided the total number of clean labels in the candidate training set is sufficiently large (throughout this paper, we refer to the dataset that consists of the examples with the associated, noisy candidate labels—as obtained in the first step of the dataset construction process—as the candidate dataset/candidate training set). This result questions the expensive practice of building clean humanly labeled training sets, and suggests that it can be better to collect larger, noisier datasets instead.

Specifically, we consider the CIFAR-10 classification problem, where the goal is to classify 32x32 color images in 10 classes. The clean training set consists of 5000 images per class and was obtained by labeling the images from the Tiny Images dataset with expert labelers. Instead of training on the original, clean training set, we train on a new noisy training set consisting of candidate examples only. We constructed this candidate training set by picking the images from the Tiny Images dataset with the labels of the CIFAR-10 classes. Only about half of the examples in this candidate dataset are correctly labeled. We quantify performance on the new CIFAR-10.1 test set from Recht et al. (2019). We ensured by automatically and manually cleaning the training set that it contains no images that are similar to any of the images in the CIFAR-10.1 test set.

We trained the best performing networks for CIFAR-10, (ResNet (He et al., 2016), Shake-Shake (Gastaldi, 2017), VGG (Simonyan & Zisserman, 2015), DenseNet (Huang et al., 2017) and others (Han et al., 2017; Liu et al., 2018)), and found that with early stopping the training we achieve significantly higher test accuracy than when training on the original, clean examples. That is possible, because even though the candidate dataset is very noisy, it contains more clean examples than the original CIFAR-10 training set, as additional experiments show.

More specifically, the best performing model trained on the original CIFAR-10 training set has 7% classification error on the CIFAR-10.1 test set (Recht et al., 2019). The best performing model trained on our candidate training set, in contrast, achieves the better error rate of only 5.85%, with early stopping training of the PyramidNet-110 model. This is a lower (better) test accuracy (as measured on the CIFAR-10.1 dataset) than that achieved by any model trained on the clean CIFAR-10 training set. What is more, all models we tested have a significant gap of training on the original clean and our noisy data. For example, the performance of ResNet-18 jumps from 85.2% to 92.15% when training and early stopping on the noisy dataset instead of the clean one.

Early stopping is critical to achieve the best performance. By keeping track of the performance of a clean subset of the noisy training set, we show that the clean labels are fitted significantly faster than the noisy labels, and a good point to stop is when most of the clean labels are fitted well by the network, because at this point most of the wrong labels have not been fitted yet.

We also study the difference between the "real" noise in the data obtained through the actual data collection progress (from sources such as search engines, like Tiny Images) and artificially generated noise through randomly flipping the labels of a clean dataset, which is often used in the literature as a proxy for the former. We show that the "real" noise is more structured and therefore it is easier to fit and less harmful to the classification performance in contrast to artificially generated noise, which is harder to fit and potentially much more adversarial to performance.

Finally, we show that our results continue to hold for other datasets. Specifically, we consider classifying invertebrate animals from training on i) a candidate dataset we collected from Flicker only through keyword search es and without subsequent labeling and from ii) a subset of the ImageNet. While for this experiment training on the candidate dataset does not improve over training on the clean set, early stopped gradient descent achieves performance close to that obtained when training on the clean set.

## 1.2 RELATED WORK

While our work focuses on *exploiting* the fact that large neural networks fit clean labels faster than noise, a large number of related works have shown that neural networks are robust to label noise, both in theory and practice (Rolnick et al., 2017), and have proposed methods to make deep networks even more robust to label noise (Sukhbaatar & Fergus, 2014; Jindal et al., 2016). Several other papers have

suggested loss function adjustments and re-weighting techniques for noise robust training (Zhang & Sabuncu, 2018; Ren et al., 2018; Tanaka et al., 2018).

Guan et al. (2018) has demonstrated that the classification performance of deep networks remains almost constant when training on partly randomly perturbed MNIST (handwritten digits) training examples. A number of recent works have offered explanations for why deep neural networks fit structure in data before fitting noise or other complex patterns: Arpit et al. (2017) has shown that loss sensitivity of clean examples is different to noisy examples, Ma et al. (2018) demonstrated that DNNs model low-dimensional subspaces that match the underlying data distribution during the early stages of training but needs to increase the dimensionality of the subspaces to fit the noise later in training, and finally Li et al. (2019); Arora et al. (2019) have attributed this in the over-parameterized case with the low-rank structure of the Jacobian at initialization. There are few works that have exploited the fact that clean examples are fitted faster than noise, with three notable exceptions being the paper by Shen & Sanghavi (2019) which achieved noise-robustness by proposing a iterative re-training scheme, Sun et al. (2018) which proposed systematic early stopping, and Song et al. (2018) which trains networks with large learning rates and uses the resulting loss to separate clean from misslabeled examples. Li et al. (2019) provided the perhaps first theoretical justification for early stopping to achieve robustness, by showing that gradient descent with early stopping is provably robust to label noise for overparameterized networks.

Next we note that, we have used the Tiny Images dataset as a source for obtaining (noisy) candidate examples; other works have used it as a source for augmenting the clean CIFAR-10 and CIFAR-100 datasets (Carmon et al., 2019; Laine & Aila, 2016) for a semi-supervised learning setting.

Finally we note that most of the works on learning from noisy labels have used artificially generated noise on clean, baseline datasets for motivation, experimentation and as performance metrics. As discussed later in this paper, we find that the performance degradation resulting from overfitting the noisy data is significantly worse on such artificially generated noise, whereas overfitting to the true label noise is not nearly as detrimental. The reason is that the true label noise found in the Tiny Images dataset is highly structured and not adversarial.

## 2 DATASET COLLECTION AND PROBLEM SETUP

We construct a candidate training dataset for the CIFAR-10 classification problem, without any explicit human labeling effort. Our noisy training set is a subset of the 80 million Tiny Images dataset (Torralba et al., 2008). The Tiny Images dataset consists of 32x32 images obtained by searching for nearly $80,000$ query keywords or labels in the image search engines Google, Flickr, and others. This yields examples of the form (keyword, image), but the associated labels (keywords) are very noisy, specifically only about 44% are correct (Torralba et al., 2008, Sec. III-B), and often an example image is completely unrelated to the label (keyword).

The goal of the CIFAR-10 classification problem is to classify 32x32 color images into the ten classes {airplane, automobile, bird, cat, deer, dog, frog, horse, ship, truck}. The original, clean, training set for the CIFAR-10 dataset was obtained by first extracting candidates from the noisy examples of the Tiny Images dataset for each of the ten classes based on the finely grained tiny-images keywords (for example, keywords for the class dog are: puppy, pekingese, chihuahua, etc., see (Recht et al., 2019, Table 6) for a complete list), and then labeling all of the so-obtained images by human experts (Krizhevsky, 2009). This gives a clean training set consisting of 5000 examples per class.

We construct a new and larger (by a factor of about 10) noisy training set by simply taking the subset of the Tiny Images for the respective class according to the same finely grained Tiny Images keywords, without any further human labeling. This resembles a data collection process without any explicit human labeling or cleaning of the data.

In our experiments throughout this paper, test performance is measured on the CIFAR-10.1 test set from Recht et al. (2019). To ensure that our noisy training set does not overlap with the CIFAR-10.1 test set in any way, we carefully removed all test images as well as images that are similar to test images from the noisy training set. Both is necessary because the extracted subset of the Tiny Images dataset contains many images that are similar to test images up to slight differences in color scale, contrast, original resolution, croppings, and watermarks. See Appendix A on details and

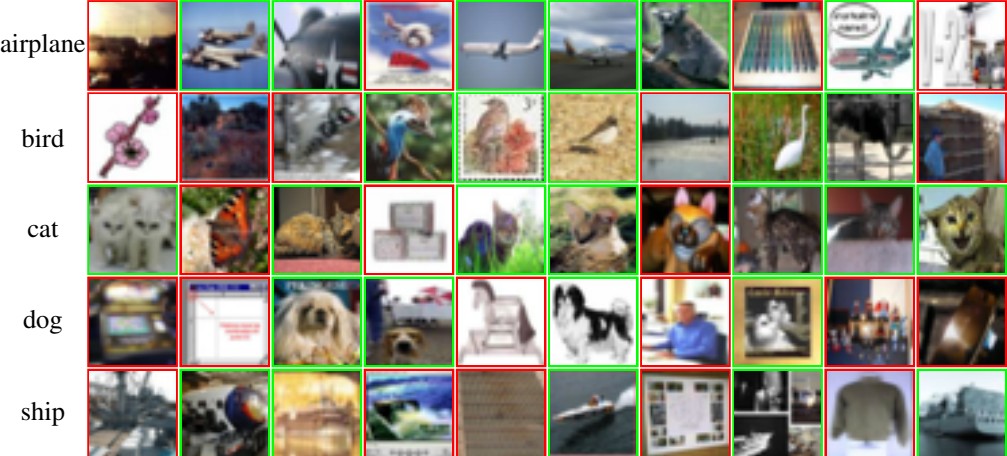

Figure 1: Randomly selected images from our noisy training set for some CIFAR-10 classes. Each row corresponds to a class and contains 10 randomly selected images. Note that about half of the images (marked with a red rectangle) are completely unrelated to the class and even to other classes.

examples of this non-trivial data cleaning process. We work with the CIFAR-10.1 test set instead of the original CIFAR-10 test set because it contains fewer images (2000 instead of 10000), and thus made our manual data cleaning process faster. Both datasets share the same distributional properties, and Recht et al. (2019, Figure 1) found the test performance on both sets to be linearly related.

After removing the similar images, our noisy training set consists of the number of examples indicated in the table below. About half of the labels in the noisy training set are wrong, see Figure 1 for an illustration. Also note that they are often completely unrelated and out of distribution—for example a screenshot, a picture of a person, and a wheel are all assigned the label "dog".

|  | plane | auto | bird | cat | deer | dog | frog | horse | ship | truck | total |
|---|---|---|---|---|---|---|---|---|---|---|---|
| our candidate training set | 39k | 53k | 35.5k | 47.5k | 38k | 48k | 33.5k | 41.5k | 46.5k | 44k | 426.5k |
| clean CIFAR training set | 5k | 5k | 5k | 5k | 5k | 5k | 5k | 5k | 5k | 5k | 50k |

## 3 TRAINING ON A LARGE NOISY DATASET CAN BE BETTER THAN TRAINING ON A CLEAN ONE

In this section, we show that the performance of a deep network trained on clean examples can be matched or even bested if the same network is trained with early stopping on noisy candidate examples—despite the many wrong labels—as long as there are sufficiently many clean labels in the noisy dataset. Our results show that this is due to neural networks fitting clean labels significantly faster than noise; specifically we observe that a subset of our training set that has clean labels is fitted during the early stages of training and much faster than the overall training set. Thus, when stopped early, the network is at a state similar to that obtained when only trained on the clean labels in the first place. This effect enables achieving classification performance by training on very noisy, potentially automatically collected candidate datasets that is on par with the performance obtained by training on very clean, expert annotated datasets.

Figure 2 depicts the training and test accuracy of training with stochactic gradient descent on our candidate dataset as well as on the clean CIFAR-10 training dataset for ResNet-18 with default parameters, taken from the Pytorch GitHub repository (PyTorch). The results show that when trained on the noisy dataset, the network achieves significantly better test performance than when trained on the clean dataset. Moreover, the best performance is achieved when the network is early stopped at about 30 epochs. At this optimal early stopping point, essentially all the clean examples but only very few of the noisy examples are fitted. This follows by noting that at about 30 epochs, the network achieves almost 100% training accuracy on a clean subset of the noisy dataset, but only about 65% overall test accuracy at a false-label-rate of about 50%. As test set we used CIFAR-10.1, and the clean subset of the training examples is the original CIFAR-10 test set, which is by construction a

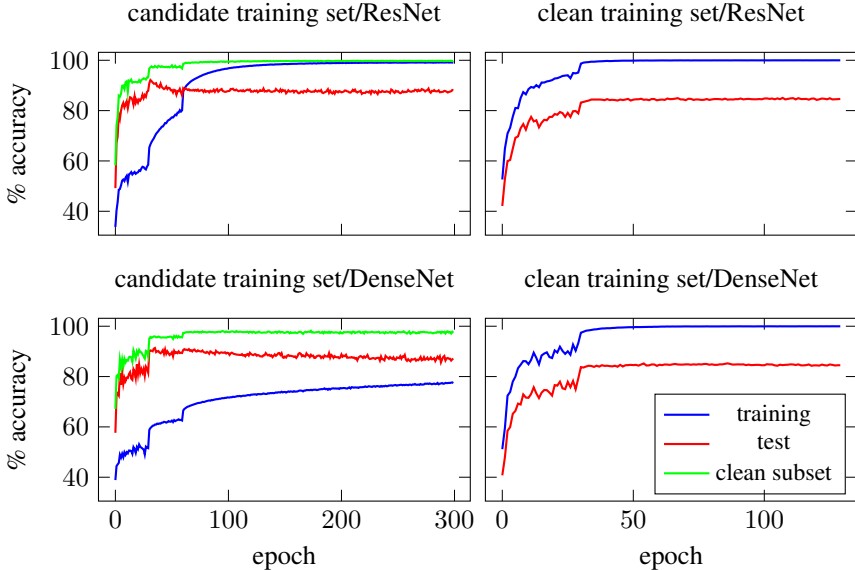

Figure 2: Accuracy on the training set, test set throughout training of the ResNet-18 and DenseNet-110 (k=12) models on our training set (left) and original CIFAR-10 training set (right). For the experiment on our training set, we also show the accuracy on a subset of the training set that is known to have clean labels and corresponds to CIFAR-10 test set.

subset of the candidate training set. In practice, the optimal stopping time can be obtained in a data driven way by monitoring the test performance on a small, clean subset of the data, and stopping once close to 100% training accuracy is achieved on the clean subset of the data.

ResNet is sufficiently overparameterized to perfectly fit the training data, even the mislabeled examples. However, even a network with fewer parameters, such as the DenseNet-110 (k=12) fits the clean data before the network reaches its capacity to fit the noisy labels and the training accuracy plateaus at about 80%, see Figure 2.

We observe this effect across a large number of standard and state-of-the-art models (see Appendix B). In Table 1, we provide the main results for the models we have tested on both the original clean CIFAR-10 training set and our candidate training set. The reported accuracies correspond to the best ones achieved during training for each setup. We note that there is a significant improvement in test performance across all models. We also observe that the number of parameters does not seem to be directly related to the performance increase of the models.

| | #Parameters | Clean training accuracy | Noisy training accuracy | Gap |
|---|---|---|---|---|
| Shake-Shake-26 2x64d | $1.19 \cdot 10^7$ | 87.6 | 93.6 | 6 |
| PyramidNet-100 ($\alpha = 270$) | $2.85 \cdot 10^7$ | 87.5 | 94.15 | 6.65 |
| DenseNet-BC-100 (k=12) | $7.69 \cdot 10^5$ | 85.25 | 91.35 | 6.1 |
| ResNet-18 | $1.12 \cdot 10^7$ | 85.2 | 92.15 | 6.95 |
| VGG-16 | $1.47 \cdot 10^7$ | 84.65 | 89.9 | 5.25 |
| PNASNet | $4.52 \cdot 10^5$ | 81.25 | 89.55 | 8.3 |

Table 1: Model accuracy (in percent) on the CIFAR-10.1 test set when trained on the original CIFAR-10 training set and our noisy training set. Number of parameters is the total trainable number of parameters of the model and gap is the difference between the corresponding accuracies of the two training sets.

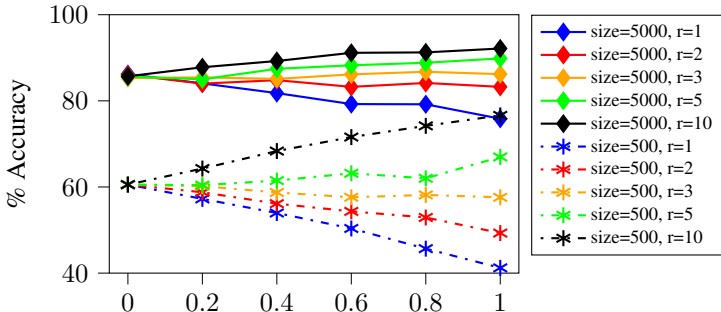

Figure 3: Average test performance of ResNet-18 for training on datasets with varying number of examples and noise level: The fraction 0 corresponds to a dataset entirely consisting of clean labels of the CIFAR-10 traing set; at fraction 0.5, half of the clean images are substituted each with $r$ many images from the noisy dataset. The results show that that if the number of clean examples grows sufficiently fast with the noisy level, the test accuracy improves.

## 4 HOW CLEAN DO THE LABELS HAVE TO BE?

In the previous section we found that learning on the noisy data gives significantly better performance than training on the clean data. Next we demonstrate that this finding relies on the total number of clean labels that are available—if that number grows sufficiently fast with the noise level, then performance becomes increasingly better when training on the noisy training set.

We perform the following experiment to measure performance on increasingly larger and noisier datasets. We start with the original CIFAR-10 training set and gradually replace a fraction of it with $r = 1, 2, 3, 5, 10$ times more images from the Tiny Images dataset. For example, if the fraction is 0.5 and $r = 2$, that means that the new, noisy dataset consists of 25.000 images of the original clean CIFAR-10 training set and of 50.000 noisy images from our noisy training set. Increasing the fraction for some fixed value of $r > 1$ thus increase both the dataset size as well as the noise level. We did this experiment both in the regime of very few examples (500 per class) and many examples (5000 per class). Figure 3 depicts the results.

For $r = 1$, the total dataset size remains constant but becomes increasingly noisier. Thus the effective number of clean examples in the noisy dataset becomes smaller as the fraction increases. Non-surprisingly, we observe that the test accuracy drops as the noise increases. The curve for $r = 5$ is more interesting: If we substitute one clean label with 5 noisy ones, then the test performance increases even though the noise level increases, because the total number of clean examples also grows sufficiently fast.

## 5 TRUE VERSUS SYNTHETIC LABEL NOISE

As mentioned in the introduction, a large number of papers has shown that neural networks fit *randomly* corrupted labels slower than clean ones and/or that neural network are robust to label noise (Zhang et al., 2017; Rolnick et al., 2017; Guan et al., 2018; Ma et al., 2018; Li et al., 2019). The experiments in those works are exclusively for noisy labels generated by randomly flipping the clean labels, which can be considered as in-distribution noise. As we see from Figure 1, the 'true' label noise generated through the image collection process is very different in that it contains lots of out-of-distribution images as well as structured noise. For example, the keyword 'Mouser' used to query images for the cat class by CIFAR-10 yields mostly toy guns and listing images of an electronics distribution company and only a few cat related images. Similarly, the keyword 'Tomcat' surprisingly yields only few cat related images as well. Most of the images are related to Java server implementations, the F-14 jet nicknamed Tomcat, and a species of insects.

Such structured, "real" noise is easier to fit and less harmful to classification performance. To demonstrate this, we take the original CIFAR-10 training set and perturb 45% of the labels by uni-

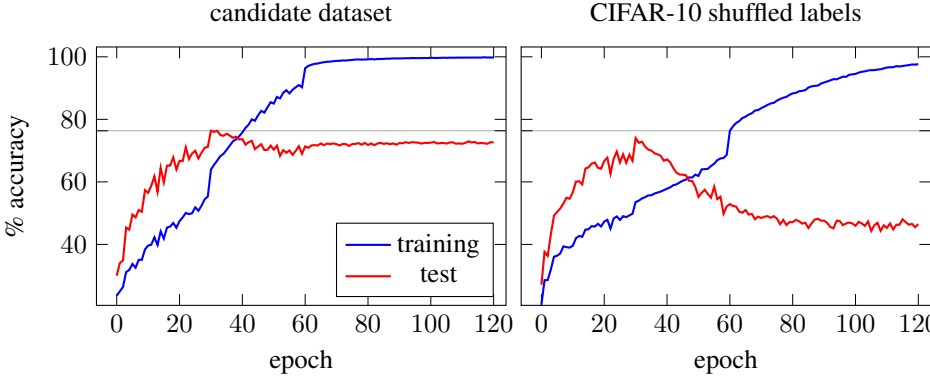

Figure 4: Accuracy on the training set and test set throughout training of the ResNet-18 model on a $50,000$ image subset of our training set (left) and original CIFAR-10 $50,000$ image training set with the labels permuted such that each class retains $55\%$ clean labels (right).

formly flipping them. We then train on the so-obtained artificially perturbed dataset and a subset of our noisy dataset consisting of the same number of examples (5000 per class). This process ensures that the label error probability is approximately the same, but the distribution of the errors is very different. The results in Figure 4 show that the data from our noisy dataset is fitted significantly faster because the noise is more structured. Moreover, the results show that early stopping is important for both setups, but training until overfitting is not nearly as detrimental to the test performance for training on the candidate dataset than on the noisy one.

## 6    EXTENSIONS TO A SUB-SET OF IMAGENET

The finding that it is possible to build a well-performing classifier by training and early stopping on a noisy candidate dataset is not unique to the CIFAR-10 classification problem. In order to demonstrate this, we train a classifier for a subset of the popular ImageNet Deng et al. (2009) dataset.

We randomly choose 15 classes out of the 61 classes belonging to the family class of *invertebrates* (any animal lacking a backbone). In order to construct a noisy candidate set, we use the synsets of these classes as search keywords to obtain candidate images from Flickr. In order to avoid any overlaps between the original ImageNet dataset and the candidate dataset, we restrict the search date to a 5-year period, from 2014-07-12 to 2019-07-12. We also search in 4-month intervals at a time and restrict the maximum number of images from each interval in order to prevent any bias that might be a result of mass uploads from institutional and professional accounts. We do not filter or further label any of the images obtained from the Flickr searches, only resize them to 256x256, which is the size of the images in the ImageNet. The resulting candidate dataset consists of around 173k images across 15 classes. Note that this is an extremly noisy dataset, the average accuracy of image search results from the internet is considered to be around 10% Torralba et al. (2008).

In Figure 5, we show the training and test accuracy of training with stochastic gradient descent on our candidate dataset obtained from Flickr as well as the subset of ImageNet training set corresponding to the chosen 15 classes for the ResNet-18 model with default parameters. The results show that when trained on the candidate dataset, the model achieves a comparable test performance to when trained on the clean dataset. Moreover, there is a clear early stopping point (about 30 epochs) during the training on the candidate dataset where the best test performance is achieved. Our manual inspections of the candidate dataset reveal that the dataset is very noisy with falsely labeled images accounting to much more than half of all the images for each class. Moreover, a large part of the falsely labeled images comes from uploads from few, but highly active accounts. This also hinders the independence of the noise distribution across different classes as there are few prominent sources of the noise, which in turn makes the noise more adversarial. This partly explains the high variance and lower test performance during the early steps of training on the candidate dataset. We note that in practice using more specific search keywords instead of the ImageNet synsets can yield much better results.

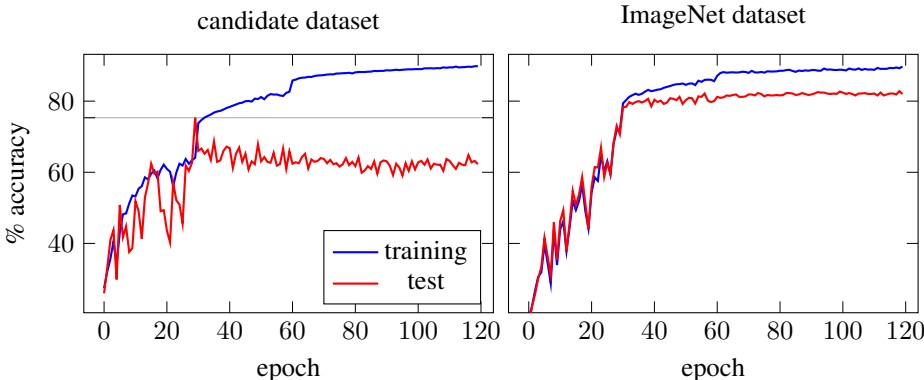

Figure 5: Accuracy on the training set, test set throughout training of the ResNet-18 model on our candidate training set (left) that was obtained from Flickr searches and original ImageNet training set (right) for the classification problem consisting of 15 children classes of the parent class corresponding to invertebrate animals.

## 7 REGULARIZATION WITH EARLY STOPPING: APPEALING TO LINEAR MODELS

For understanding why highly overparameterized neural networks fit structure faster than noise, it is helpful to appeal to linear models. Specifically, a number of recent works show that highly overparameterized models behave like an associated linear model around the initialization, which in turn offers an explanation why neural networks fit structure faster than no structure (Li et al., 2019; Du et al., 2019; 2018; Arora et al., 2019). While the regime where those results hold is extremely overparameterized and thus might not be the regime in which those models actually operate (Chizat et al., 2018), it can provide some intuition on the dynamics of fitting clean versus noisy data. In this section, we first briefly comment on those recent theoretical results and explain how they provide intuition on why deep networks fit structured data faster than noise, and second, inspired by those explanations for linear models, we demonstrate that early stopping the training of non-deep overparameterized *linear* models enables learning from noisy examples as well.

### 7.1 EXISTING THEORY FOR OVERPARAMETERIZED MODELS

Consider a binary classification problem, and let $f_{\mathbf{W}} \colon \mathbb{R}^d \to \mathbb{R}$ be a neural network with weights $\mathbf{W} \in \mathbb{R}^N$, mapping a $d$-dimensional features space to a label. The output label associated with example $\mathbf{x}$ is interpreted as belonging to class 1 if $f_{\mathbf{W}}(\mathbf{x})$ is positive and as belonging to class $-1$ if $f_{\mathbf{W}}(\mathbf{x})$ is negative. Let $\{(y_1, \mathbf{x}_1), \ldots, (y_n, \mathbf{x}_n)\}$ be a set of $n$ many training examples. The network is trained by minimizing the empirical loss

$$\mathcal{L}(\mathbf{W}) = \frac{1}{2}\|\mathbf{y} - \mathbf{f}_{\mathbf{W}}(\mathbf{X})\|^2,$$

with gradient descent, where $\mathbf{y} = [y_1, \ldots, y_n]$ are the labels of the examples and $\mathbf{f}_{\mathbf{W}}(\mathbf{X}) = [f_{\mathbf{W}}(\mathbf{x}_1), \ldots, f_{\mathbf{W}}(\mathbf{x}_n)]$ are the predictions of the model. Let $\mathcal{J}(\mathbf{W}) \in \mathbb{R}^{n \times N}$ be the Jacobian of the mapping from weights to outputs $\mathbf{f} \colon \mathbb{R}^N \to \mathbb{R}^n$, and let $\mathbf{J} \in \mathbb{R}^{n \times N}$ be a specific matrix that obeys $\mathbf{J}\mathbf{J}^T = \mathbb{E}\left[\mathcal{J}(\mathbf{W}_0)\mathcal{J}^T(\mathbf{W}_0)\right]$, where expectation is with respect to the random initialization $\mathbf{W}_0$. For two-layer networks, a number of very recent works (Li et al., 2019; Du et al., 2019; 2018; Arora et al., 2019) have shown that gradient descent in close proximity around the initialization is well approximated by gradient descent applied to the associated linear loss

$$\mathcal{L}_{\text{linear}}(\mathbf{W}) = \frac{1}{2}\|\mathbf{f}_{\mathbf{W}_0}(\mathbf{X}) + \mathbf{J}(\mathbf{W} - \mathbf{W}_0) - \mathbf{y}\|^2.$$

From this, it follows that (see (Li et al., 2019; Arora et al., 2019)) that the prediction error after $t$ iteration of gradient descent with stepsize $\eta$ (provided that $\eta t$ is not too large) is given by

$$\|\mathbf{y} - \mathbf{f}_{\mathbf{W}_t}(\mathbf{X})\|^2 \approx \sum_{i=1}^{n}(1 - \eta\sigma_i^2)^{2t}(\langle \mathbf{y}, \mathbf{v}_i\rangle)^2, \tag{1}$$

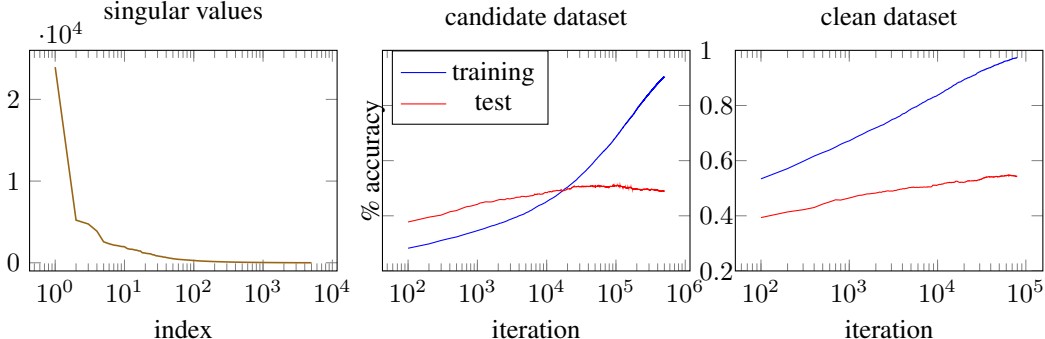

Figure 6: Training a large random feature model on a subset of our candidate training set (middle) and the CIFAR-10 training set (right) with gradient descent. Singular values of the feature matrix (left) shows the low-rank structure for our candidate training set.

where $\mathbf{v}_i$ are the eigenvectors and $\sigma_i^2$ are the eigenvalues of the matrix $\mathbf{J}\mathbf{J}^T$, and $\eta \leq 1/\sigma_{\max}^2$. Equation (1) shows that the directions associated with large eigenvectors are fitted significantly faster by gradient descent than directions associated with small eigenvalues. Thus labels vectors $\mathbf{y}$ that are well aligned with eigenvectors corresponding to large eigenvalues are fitted significantly faster than those corresponding to small eigenvalues. The eigenvalues and eigenvectors are determined by the structure of the network. It turns out that for neural network the structure is such that the projection of correctly labeled data (i.e., $\langle \mathbf{v}_i, \mathbf{y} \rangle$, for $\mathbf{y}$ a set of correct labels) is much more aligned with directions associated with large eigenvalues of $\mathbf{J}\mathbf{J}^T$, than the projection of falsely labeled data (i.e., $\langle \mathbf{v}_i, \mathbf{y} \rangle$, for $\mathbf{y}$ a set of falsely labeled examples). This suggests that for highly linear overparameterized models we expect to see similar dynamics in that for certain model structure is fitted faster than noise—a property that we demonstrate in the next section.

## 7.2 NON-DEEP MODELS

Finally, we demonstrate that even with non-deep overparameterized models, clean labels are learned faster than noisy ones, and thus early stopping enables learning from (noisy) candidate examples. We consider a random feature model fitted with gradient descent and early stopped. The reason for considering a random feature model is that such models can reach performances that are reasonably close to that of a neural network if the number of random features, and thus the size of the model, is sufficiently large.

We consider a random-feature model as proposed in Coates et al. (2011). We first extract for each example random features through a one-layer convolutional network $G$ with $4000$ random filters, each of size $6 \times 6$. For each image $\mathbf{x}$, this gives a feature vector $G(\mathbf{x})$ of size $m = 72000$. Consider a training set with $N$ examples, and let $\mathbf{X} \in \mathbb{R}^{N \times m}$ be the feature matrix associated with the training set, and let $\mathbf{Y} \in \mathbb{R}^{N \times K}$ be the matrix containing the labels. Specifically, $\mathbf{Y}$ has a one in the $k$-th position of the $i$-th row if the $i$-th training example belongs to the $k$-th class, $k \in \{1, \ldots, K\}$. In the training phase, we fit a linear model by minimizing the least-squares loss

$$\mathcal{L}(\mathbf{Z}) = \|\mathbf{Y} - \mathbf{X}\mathbf{Z}\|_F^2,$$

with gradient descent. Here, the matrix $\mathbf{Z} \in \mathbb{R}^{m \times K}$ specifies the model, along with the random featurizer (i.e., the random convolutional model). Given a new example $\mathbf{x}$, the model then assigns the label $\arg\max_k G(\mathbf{x})\mathbf{Z}$.

We trained the model above on two datasets: a clean dataset consisting of 500 examples for each class of the CIFAR-10 training set, and a candidate dataset consisting of 1500 examples for each class from our candidate dataset. We measure classification performance as done throughout on the CIFAR-10.1 test set. The results, shown in Figure 6, show that on the clean dataset, this model achieves essentially 100% training accuracy. On the noisy dataset, similarly as for the deep networks, early stopping improves performance in that the best performance is achieved at about 70,000 iterations of gradient descent, long before the best fit of the model is achieved.

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

# A   DETAILS ON THE DATA-CLEANING PROCESS

In order to make sure that the CIFAR-10.1 test set has no overlap with our noisy dataset, we constructed our noisy dataset by first extracting all images with the respective keywords from the Tiny Images dataset, and then removing all images that are similar to the images in the CIFAR-10.1 test set as follows: We first identified 100 closest images for each individual image in the CIFAR-10.1 test set in $\ell_2$-distance, and then removed equivalent and similar images after manual visual inspection of these candidates.

We consider two images similar, if they share the whole or a considerable portion of the same underlying main object, secondary object, or a specific background (see Figure 7 for examples). Performing this step manually is necessary because even images that are relatively far in $\ell_2$-distance are often extremely similar, while images relatively close can be significantly different (see Figure 8 for examples). We also note that it is too pessimistic to cutoff the images by using a threshold together with a closeness measure because of this wide variety of differences across images that actually contain the same object. Figure 8 demonstrates an instance of this problem for an image in the 'bird' class. There are many other such instances with highly different distance values, which necessitates manual inspection for this part.

In the following figures (9–11), we showcase some instances that further demonstrate the need for expert labeling for this similar image identification task. We note that in addition to the provided similar images, all the test images that are shown here also have identical copies in our training set that can be found without any need for manual labor. This is expected because the CIFAR-10.1 test set is compiled mostly from Tiny Images and it holds true for most of the images in this set.

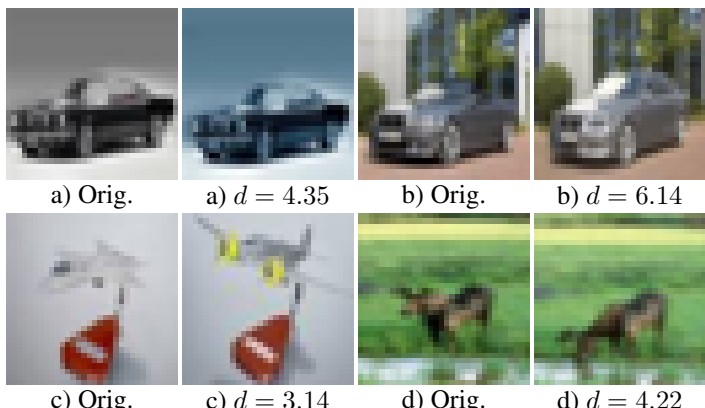

| a) Orig. | a) $d = 4.35$ | b) Orig. | b) $d = 6.14$ |
| c) Orig. | c) $d = 3.14$ | d) Orig. | d) $d = 4.22$ |

Figure 7: Examples of images that are considered similar for the following reasons: a) Same primary object with different contrasts; b) Different primary objects in the same background c) Same secondary object with different primary objects d) Same object in different poses.

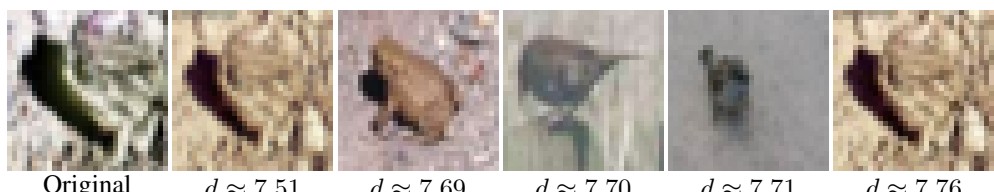

| Original | $d \approx 7.51$ | $d \approx 7.69$ | $d \approx 7.70$ | $d \approx 7.71$ | $d \approx 7.76$ |

Figure 8: Top-5 closest images to an test image in $\ell_2$-distance (d). The image at distance $d = 7.76$ is almost equal to the original one, while a closer one, at $d = 7.69$ is a different image.

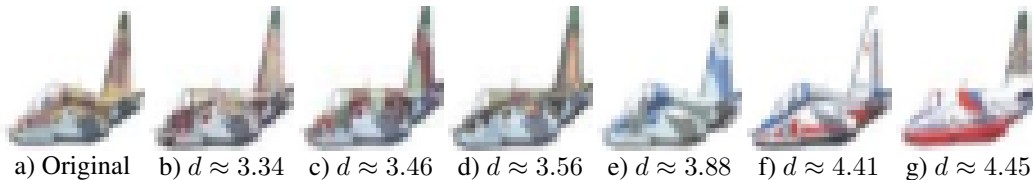

a) Original    b) $d \approx 3.34$    c) $d \approx 3.46$    d) $d \approx 3.56$    e) $d \approx 3.88$    f) $d \approx 4.41$    g) $d \approx 4.45$

Figure 9: Images that contain the same object in different colors, and are all removed from the training set.

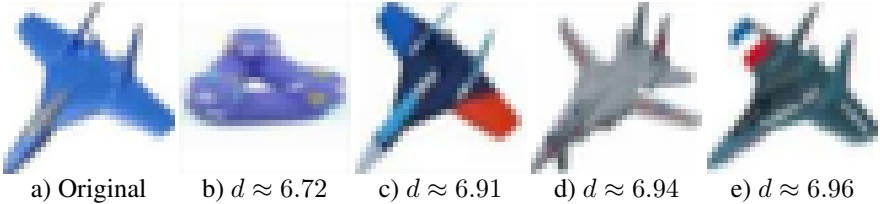

a) Original    b) $d \approx 6.72$    c) $d \approx 6.91$    d) $d \approx 6.94$    e) $d \approx 6.96$

Figure 10: Images that contain the same object in different colors as well as a different object. This demonstrates that images in different colors can have a larger $\ell_2$-distance than completely different images, thus we cannot simply remove images based on thresholding.

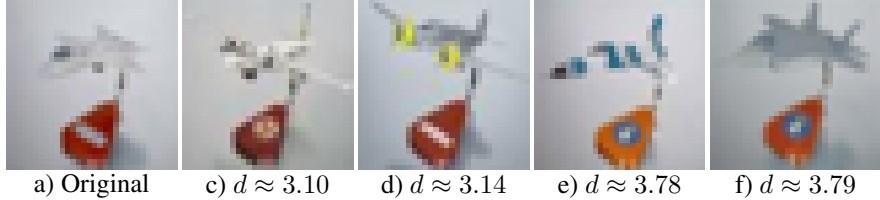

a) Original    c) $d \approx 3.10$    d) $d \approx 3.14$    e) $d \approx 3.78$    f) $d \approx 3.79$

Figure 11: Images that contain different main objects in addition to the same or very similar secondary objects in the background.

## B    LEARNING CURVES FOR DEEP LEARNING MODELS

In this section, we display additional learning curves for the models omitted in Section 3 of our main document.

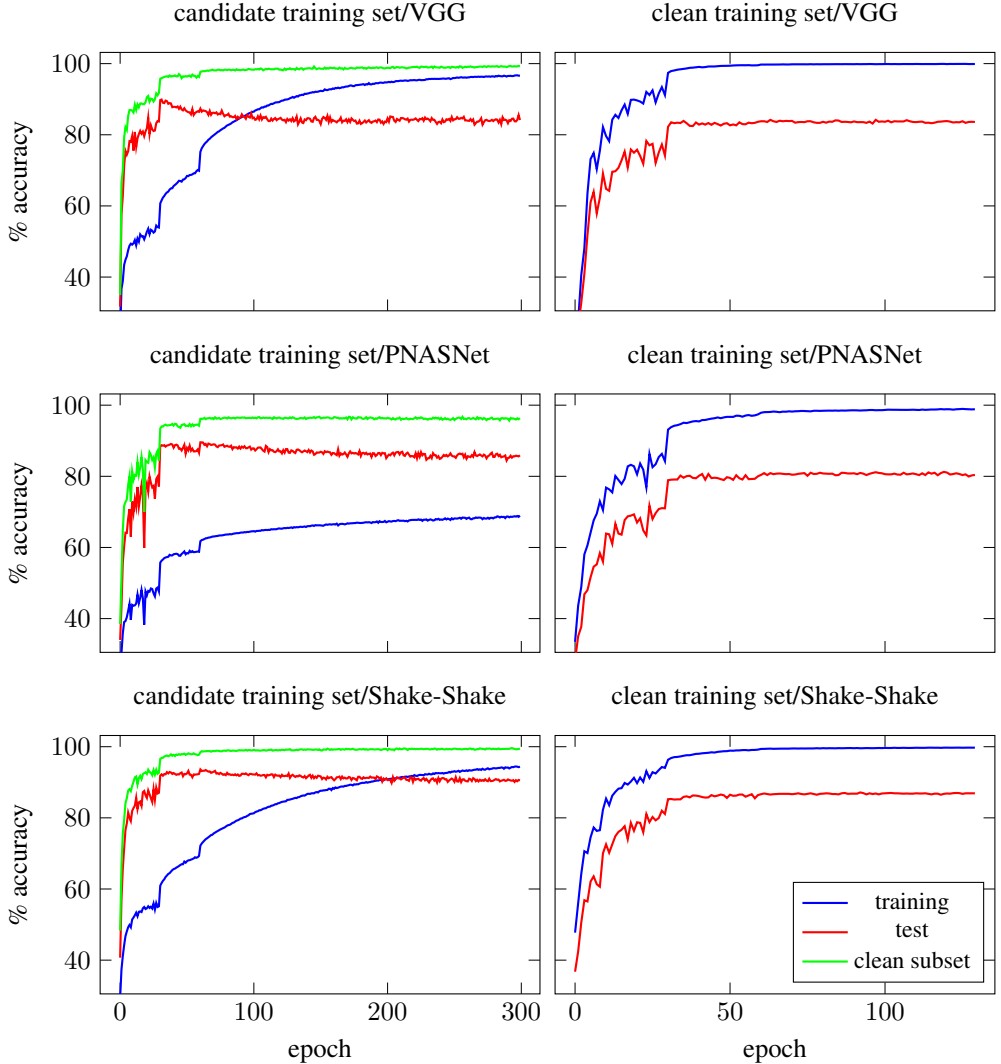

Figure 12: Accuracy on the training set, test set throughout training of the VGG-16, PNASNet, and Shake-Shake-26 2x64d (S-S-I) models on our training set (left) and original CIFAR-10 training set (right). For the experiment on our training set, we also show the accuracy on a subset of the training set that are known to have clean labels and corresponds to CIFAR-10 test set.

## C    ERROR ANALYSIS

In this section we take a closer look at the accuracy obtained by training on the candidate dataset. Figure 13 shows the accuracy per class for training ResNet-18 on the original CIFAR-10 training set and training on the candidate images with early stopping. The plots show that the model trained on the candidate examples has better performance in each class than the model trained on the original training set.

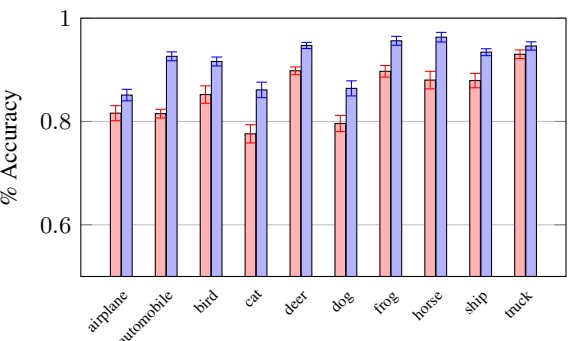

Figure 13: Accuracy per class for training ResNet18 on the original CIFAR-10 training set (red bars) and training on the candidate images with early stopping (blue bars). The plots show that the model trained on the candidate examples has better performance in each class than the model trained on the original training set. The error bars correspond to one standard deviation obtained by averaging over many different initialization.

