# OpenReview forum: "Leveraging inductive bias of neural networks for learning without explicit human annotations"
_ICLR.cc/2020/Conference — Reject_

### Official Review · AnonReviewer3 · 2019-10-23
**Official Blind Review #3**

**Rating:** 6

**Review:**

Summary

The paper exploits the training dynamics of deep nets: gradient descent fits clean labels much faster than the noisy ones, therefore early stopping resembles training on the clean labels. The paper shows that early stopping the training on datasets with noisy labels can achieve classification performance higher than when training on clean labels (on condition that the total number of clean labels in the noisy training set is sufficiently large).

The paper also makes the point that noise introduced during data collection are different from artificially generated noise through randomly flipping the labels of a clean dataset. The latter is often done in the literature. The “real” noise is more structured and therefore it is easier to fit and less harmful to the classification performance.

Strengths

The paper is well written. The results are very interesting even though they are very intuitive and simple.

Weaknesses

The idea seems to be known. For example,
Better Generalization with On-the-fly Dataset Denoising
https://openreview.net/forum?id=HyGDdsCcFQ

The paper talks about early stopping. As shown by the above paper, it is also a function of the learning rate. Please comment on what happens in the case of small learning rate and early stopping.

It would have been great to prove the theorem for deep learning. The result is limited to linear models with large number of random features.

The paper does not make it clear under what conditions early stopping prevents the model from memorizing bad labels.

The paper focuses on classification. Will the claims hold for other task types such as object detection, segmentation, etc?


**Experience Assessment:**

I have read many papers in this area.

**Review Assessment: Checking Correctness Of Derivations And Theory:**

I carefully checked the derivations and theory.

**Review Assessment: Checking Correctness Of Experiments:**

I carefully checked the experiments.

**Review Assessment: Thoroughness In Paper Reading:**

I read the paper thoroughly.

---

> ### Author Response · Authors · 2019-11-15
> **Response to review 3**
>
> On the idea seems to be known/on the fly denoising:
> The very interesting paper Song et al. ```  `  Better generalization with on-the-fly dataset denoising'' considers a similar problem (learning in the presence of noisy labels) but studies another technique. Specifically, Song et al. train a residual network with a large learning rate and use the resulting losses to separate the clean examples from mislabeled ones, based on some identifying examples that exceed a threshold. Our work, in contrast, does not attempt to separate clean labels from noisy ones but instead simply trains the network on the noisy examples.
>
> Function of learning rates: In order to address this, we performed additional experiments on ResNet18 with smaller and larger learning rates, relative to the learning rate we choose initially. Initially, we choose the default learning rate/schedule which starts at 0.1 and decays every 30 epochs. We did experiments starting from 0.05 and 0.02 and found that it changes the rate at which the training set is fitted, but not the general behavior of the curves, performance, or necessity of early stopping.
>
> Regarding under what conditions early stopping prevents memorizing bad labels: Our numerical results show that standard neural networks such as ResNet, Shake-Shake, VGG, DenseNet, and others do not memorize the data if trained for a few epochs (see Fig. 2, showing that after a few epochs only the clean labels are fitted, but not the noisy ones). We do, however, not have formal conditions stating under which conditions data is not memorized, and we are also not aware of such conditions in the literature.
>
> Regarding whether the claims hold beyond classification, e.g., for object detection or segmentation: We do not know the extent to which this claim continues to holds, but suspect that the general insight that the network imposes sufficient structure to learn from noisy labels continues to hold for other domains/problems. A concrete example where this is true is image denoising: As first shown by Ulyanov et al. ``  `  deep image prior'', a convolutional network fits a natural image faster than noise and thus enables denoising by early stopping. This is conceptually similar to what we find albeit for a different problem (image denoising) vs. learning from noisy labels.

---

### Official Review · AnonReviewer4 · 2019-10-29
**Official Blind Review #4**

**Rating:** 3

**Review:**

The paper reads well, the topic is interesting, and the connection between early stopping and fitting true label distributions before noisy ones, if generally true, is of potential impact. I do not feel the paper is ready for publication, though:

Missing context/references:
	⁃	Noisily collected labels are standard elsewhere, e.g., in sentiment analysis (self-ratings), in discourse parsing (explicit discourse markers), in weakly supervised POS tagging (crowdsourced dictionaries), in NER (Wikidata links), and in machine translation from mined parallel corpora or comparable corpora.
	⁃	Learning a ground truth from a population of turkers (MACE and subsequent work). It is well-known that the noise in such silver standard or non-adjudicated annotations is also “more structured” and to a large extent predictable (Plank et al., EACL 2014 and subsequent work on learning/predicting inter-annotator disagreements).
	⁃	Connection between L2 and early stopping. Can you replicate your results with simple L2 regularisation?
	⁃	Cost-sensitive learning of systematic disagreements (use class coherence scores for cost-sensitive learning).
	⁃	The observation that “clean examples are fitted faster than noise” is obviously related to baby steps training regimes (training on easy examples first), including active learning.

Experimental details and flaws:
	⁃	Using a black box search engine to collect labels is problematic for a few reasons: (a) Search engines change, so results are hard to reproduce. (b) Search engines are biased toward certain types of categories. Such biases will be reinforced by any model trained on this data.
	⁃	I’m always a little worried about papers that only report results for a single dataset.
	⁃	I would have liked to see some more error analysis. Are the improvements on classes with more or less support in the original dataset, for example?
	⁃	It seems to be that it would have been relatively straight-forward to construct synthetic datasets that would more directly evaluate the hypothesis that the true labels are learned first. I realize you’re interested in “real” noise rather than “random” noise - but synthetic noise doesn’t have to be random.
	⁃	That “the noise generated through the label collection process is not nearly as adversarial for learning as the noise generated by randomly flipping labels” is no surprise, and I would not present this as a finding. It would be interesting to describe the bias, e.g., by presenting confusion matrices (see Plank et al., EACL 2014).


**Experience Assessment:**

I have published one or two papers in this area.

**Review Assessment: Checking Correctness Of Derivations And Theory:**

N/A

**Review Assessment: Checking Correctness Of Experiments:**

I assessed the sensibility of the experiments.

**Review Assessment: Thoroughness In Paper Reading:**

I read the paper at least twice and used my best judgement in assessing the paper.

---

> ### Author Response · Authors · 2019-11-15
> **Response to review 4**
>
> Thanks for the review and for acknowledging that the connection between early stopping and fitting true label distributions before noisy ones is of potential impact. In our revision, we provide experiments on another dataset in order to further justify our findings and to address the reviewers concerns.
>
> Regarding context/references:
> - We fully agree that noisily collected labels are common for many problems other than image classification. However, the focus of our paper is image classification, and we thus concentrate on classification problems related to the widely popular CIFAR-10 and ImageNet classification problems. For the CIFAR-10 classification problem, it has not shown before that learning from candidate examples only is possible.
> - Thanks for bringing the paper by Plank, Hovy, and Sogaard to our attention. While a very interesting work, we do not see a connection to our work since Plank, Hovy, and Sogaard focus on NLP and incorporating measured inter-annotator agreement in the loss function. In contrast, we consider image classification and leverage the inductive bias of neural networks to learn from noisy candidate labels by early stopping. There is no mention of fitting examples slow or fast in the Plank, Hovy, and Sogaard paper.
> - Yes, we have also performed experiments with l2 regularization, but without early stopping cannot replicate our results with simply L2 regularization. Please note that our results rely on convolutional networks fitting clean labels faster than noisy ones, so early stopping is a natural way to exploit this property, while L2 regularization is not.
> - On cost-sensitive learning of systematic disagreements: This is a topic of the Plank, Howey, and Sogaard paper but not of our paper, since in our paper we do have noisy data, but no disagreements between labelers (because our data is not labeled by explicit labelers).
> - The observation that clean examples are fitted faster than noise is not related to active learning.
>
> Experimental details:
> - We fully agree with the downsides of using a search engine to collect labels pointed out by the reviewer. However, both the collection of CIFAR and ImageNet, the arguably most accepted datasets for image classification, are based on using search engines for data collection. The reproducibility of our results is not compromised by using the TinyImages dataset, because this dataset is publicly available and because we made our code available.
> Regarding reporting results only for a single dataset: In order to show that our findings extend to other datasets, we have performed an additional experiment on a subset of ImageNet and a candidate dataset that we have collected on Flickr. The results, reported in our revised paper, confirm our finding that early stopping enables learning from a candidate data set.
> - Regarding error analysis: we have added a section in the appendix showing the accuracy per class for both i) training on the candidate dataset and ii) the original CIFAR training set. The results show that the model trained on the candidate examples has a higher accuracy for every single class compared to the model trained on the original CIFAR-10 training set.
> - We did perform an experiment on a synthetic noisy dataset that directly evaluates the hypothesis that true labels are fitted faster than noise, please see Figure 4, right panel. There, we have flipped about half of the labels and the results show that the clean labels are fitted faster than the noisy ones (otherwise the test accuracy would not go first up and then drop significantly). We would also like to point out that our objective is to skip the expensive labeling step by directly training on the candidate labels. The real noise is a natural bi-product of such constructed datasets.
> - Please note that we cannot present confusion matrices as in the aforementioned paper by Plank, Hovy, and Sogaard, since we have a different setup and error mode: Our error mode is that an image does not belong to the class it is labeled with, but also not to one of the other classes (see Section 5 where we explain this).

---

### Decision · Program_Chairs · 2019-12-19

**Decision:**

Reject

**Comment:**

The authors present an approach to learning from noisy labels. The reviews were mixed and several issues remain unresolved. I do not accept the following as a valid response: "We fully agree that noisily collected labels are common for many problems other than image classification. However, the focus of our paper is image classification, and we thus concentrate on classification problems related to the widely popular CIFAR-10 and ImageNet classification problems." ICLR is a conference on theoretical and applied ML, and the fact that a technique has not been used for image classification before, does not mean you bring something to the table by doing so. The NLP literature is abundant with interesting work on label noise and should obviously be considered related work. That said, there's also missing references directly related to the connection between early stopping/regularization and label bias correction, including:

[0] https://arxiv.org/pdf/1904.11238.pdf
[1] https://arxiv.org/pdf/1705.03419.pdf
[2] http://proceedings.mlr.press/v80/ma18d/ma18d.pdf

See also this paper submitted to this conference: https://openreview.net/forum?id=SJldu6EtDS